# Human Keratinocytes Inhibit CD4^+^ T-Cell Proliferation through TGFB1 Secretion and Surface Expression of HLA-G1 and PD-L1 Immune Checkpoints

**DOI:** 10.3390/cells10061438

**Published:** 2021-06-08

**Authors:** Guillaume Mestrallet, Frédéric Auvré, Chantal Schenowitz, Edgardo D. Carosella, Joel LeMaoult, Michèle T. Martin, Nathalie Rouas-Freiss, Nicolas O. Fortunel

**Affiliations:** 1CEA, Laboratory of Genomics and Radiobiology of Keratinopoiesis, Institute of Cellular and Molecular Radiobiology, Francois Jacob Institute of Biology, DRF, 91000 Evry, France; guillaume.mestrallet@cea.fr (G.M.); frederic.auvre@cea.fr (F.A.); 2Université Paris-Saclay, 91190 Saint-Aubin, France; 3CEA, DRF, Francois Jacob Institute of Biology, Hemato-Immunology Research Department, Saint-Louis Hospital, 75010 Paris, France; chantal.schenowitz@cea.fr (C.S.); edgardo.carosella@cea.fr (E.D.C.); 4U976 HIPI Unit, IRSL, Université Paris, 75010 Paris, France

**Keywords:** skin immunity, human keratinocytes, immunomodulation, HLA-G1, PD-L1, TGFB1, tolerance

## Abstract

Human skin protects the body against infection and injury. This protection involves immune and epithelial cells, but their interactions remain largely unknown. Here, we show that cultured epidermal keratinocytes inhibit allogenic CD4^+^ T-cell proliferation under both normal and inflammatory conditions. Inhibition occurs through the secretion of soluble factors, including TGFB1 and the cell-surface expression of HLA-G1 and PD-L1 immune checkpoints. For the first time, we here describe the expression of the HLA-G1 protein in healthy human skin and its role in keratinocyte-driven tissue immunomodulation. The overexpression of HLA-G1 with an inducible vector increased the immunosuppressive properties of keratinocytes, opening up perspectives for their use in allogeneic settings for cell therapy.

## 1. Introduction

Skin grafting can be necessary in various pathological conditions, such as burns [1], traumatic injury or ulcers (Buruli ulcer, diabetes, pressure, venous ulcer), postoperatively, postinfection (necrotizing fasciitis, purpura fulminans), or in genetic diseases such as junctional epidermolysis bullosa [2].

Keratinocytes constitute the main cellular component of the epidermis. In bioengineered skin-substitute transplantation to replace large skin areas, they are obtained from biopsies of a few square centimeters in healthy body areas that are known to heal well, with minimal genotoxic exposure. The keratinocytes are then amplified ex vivo for 1 to 3 weeks, and used to generate epithelial sheets fixed on gauze, which are then deposited on the wounded area [3,4]. The current gold-standard keratinocyte amplification process is based on Green’s pioneering work [5], involving the principle of co-culturing keratinocytes and murine growth-arrested feeder fibroblasts. There have been several clinical studies on ex vivo keratinocyte amplification for skin autografts. One reported 65% transplant uptake in 63 burn patients, with 84% patient survival at 7 days [6]. Another reported 72.7% transplant uptake in 88 burn patients, with 91% patient survival [7].

However, when 40–90% of the skin surface is burned, ex vivo keratinocyte amplification is time-consuming and is variable in both quantitative and qualitative terms. Moreover, in some elderly patients or significantly inflammatory contexts, poor healing capacity can complexify clinical procedures that involve harvesting autologous skin biopsies. For these reasons, it would be useful to have banks of well-documented allogeneic bioengineered grafts as alternative materials. However, this approach is limited by the problem of transplant rejection. In allogeneic skin grafts, strong rejection is observed after 7 to 10 days. Immunosuppressants have been used to treat burn patients: methylprednisolone, cyclosporine, prednisone, anti-thymocyte globulin and azathioprine [8,9]. However, it is difficult to draw any conclusion from these studies, due to the very small numbers of patients involved. The use of immunosuppressants is also problematic because the donor site is left more susceptible to infection, and the treatment may have toxic effects on other organs such as the kidneys. It is therefore important to study the role of immune checkpoint molecules in the skin, so as to open up new approaches for reducing allograft rejection.

The overexpression of immune checkpoint molecules such as CTLA4-Ig and PD-L1 in human embryonic stem cells (hESCs) was shown to limit their rejection in humanized murine models [10]. The PD-L1 immune checkpoint molecule binds to the PD-1 protein expressed on the surface of T-cells, inhibiting their activity [11] and decreasing the autoimmune reaction [12]. HLA-G is another immune checkpoint molecule, originally described as enabling maternal-fetal tolerance [13,14]. HLA-G inhibits immune-cell functions such as the cytolytic function of NK- and T-cells, the alloproliferative response of CD4^+^ T-cells, antibody production by B-cells and the antigen-presenting function of dendritic cells [15,16]. Notably, patients expressing HLA-G were significantly less prone to acute and chronic rejection in solid-organ transplantation (heart, kidney, liver or lung) [17,18,19,20,21]. Tolerance induction by HLA-G was also demonstrated in murine models [22,23,24,25]. The HLA-G primary transcript is alternately spliced, giving rise to seven spliced transcripts that lead to seven protein isoforms. The HLA-G1 to G4 isoforms are membrane-bound, while the HLA-G5 to G7 isoforms are their soluble counterparts. The HLA-G1 protein is the full-length isoform, with three extracellular domains (α1, α2, and α3), and it is also the main HLA-G isoform expressed in vivo [26].

Several approaches to using membrane-bound or soluble HLA-G forms as immunoregulatory agents in transplantation have been documented. The first was to use epithelial cells derived from HLA-G^+^ amniotic cells. These epithelial cells express HLA-G but not HLA-DR or the co-stimulation molecules CD40, CD80 and CD86. They are also able to limit peripheral blood mononuclear cell (PBMC) alloproliferation [26]. The second approach was to overexpress HLA-G in embryonic stem cells and their epidermal derivatives, reducing lysis by NK cells and inhibiting T-cell alloproliferation in mixed lymphocyte reactions [27]. The third approach was to use the soluble form of HLA-G to inhibit transplant rejection. Notably, the expression of the HLA-G tetramer in transgenic mice reduced acute skin allograft rejection [22]. Additionally, various recombinant HLA-G proteins enhanced tolerance to skin allografting in mice [23]. Taken together, these results identify HLA-G as a promising molecule to limit skin allograft rejection.

The present ex vivo study reports the ability of both non-cultured tissue-extracted keratinocytes and keratinocytes amplified in culture to reduce CD4^+^ T-cell proliferation, and proposes approaches to boost these immunomodulatory properties.

## 2. Materials and Methods

### 2.1. Human Tissues and Cells

The present study was approved by the review board of the iRCM (Institut de Radiobiologie Cellulaire et Moléculaire, CEA (Atomic Energy Commission), Fontenay-aux-Roses, France), and is in accordance with the scientific, ethical, safety and publication policy of the CEA (CODECO number DC-2008-228, reviewed by the ethical research committee IDF-3). PBMCs were collected from healthy donors in the French Blood Bank (EFS) at Saint Louis Hospital, Paris, after informed consent was obtained. Human skin tissue from healthy adult donors was collected in the context of breast reduction surgery, after obtaining informed consent. Epidermal keratinocytes and dermal fibroblasts were extracted as previously described [28]. Briefly, enzymatic treatment with a solution containing (*v*/*v*) 3/4 grade II dispase 2.4 U mL^−1^ (Roche Molecular Biochemicals, Mannheim, Germany) and 1/4 trypsin 0.25% (Gibco) was conducted for 24 h at 4 °C. We used two categories of cells in this study: cells directly extracted from the tissue and not amplified (tissue keratinocytes), and cells extracted from the tissue, amplified and used between passage 1 and 3 in culture (amplified keratinocytes).

### 2.2. Cell Culture

Amplified adult epidermal keratinocytes were obtained as follows: bulk cultures were generated in a serum-containing medium, in the presence of a feeder layer of human dermal fibroblasts growth-arrested by 60 Gy γ irradiation [29]. All cultures were performed in plastic flasks coated with type-I collagen (BioCoat, Becton-Dickinson, Le Pont de Claix, France). The composition of the serum-containing medium consisted of DMEM and Ham’s F12 media (Gibco, ThermoFisher, Les Ulis, France) (*v*/*v*, 3/1 mixture), 10% fetal calf serum (Hyclone, Fisher Scientific, Illkirch, France), 10 ng/mL epidermal growth factor (EGF) (Chemicon, Fisher Scientific, Illkirch, France), 5 μg/mL transferrin (Sigma, Saint-Quentin Fallavier, France), 5 μg/mL insulin (Sigma, Saint-Quentin Fallavier, France), 0.4 μg/mL hydrocortisone (Sigma, Saint-Quentin Fallavier, France), 180 μM adenine (Sigma, Saint-Quentin Fallavier, France), 2 mM tri-iodothyronine (Sigma, Saint-Quentin Fallavier, France), 2 mM L-glutamine (Gibco, ThermoFisher, Les Ulis, France), and 100 U/mL penicillin/streptomycin (Gibco, ThermoFisher, Les Ulis, France). The medium was renewed 3 times a week. For cell amplification, keratinocytes were seeded at 1000 cells/cm^2^ and sub-cultured weekly. Feeder cells were seeded at 5000 cells/cm^2^. In all experiments, the influence of inflammatory conditions was analyzed through pretreatment with interferon-γ (IFN-γ) (Peprotech, Neuilly-sur-Seine, France) and tumor necrosis factor-α (TNF-α) (R&D Systems, Bio-Techne, Noyal Châtillon sur Seiche, France).

### 2.3. Flow Cytometry Analysis

For analysis of cell-surface immune marker expression, keratinocytes were processed as single-cell suspensions and stained for 1 h at room temperature with monoclonal antibodies. The staining antibodies used were as follows: PE-cy7-conjugated mouse anti-human PD-L1 monoclonal antibody (clone MIH1, Thermo Fisher, Les Ulis, France), FITC-conjugated mouse anti-human CD40 monoclonal antibody (clone HB14, Thermo Fisher, Les Ulis, France), FITC-conjugated mouse anti-human MHC1 monoclonal antibody (clone W6-32, Thermo Fisher, Les Ulis, France), Alexa 700-conjugated mouse anti-human HLA-G monoclonal antibody (clone 87G, Novus Biologicals, Lille, France), PE-conjugated rat anti-human ILT4 monoclonal antibody (clone 42D1, Thermo Fisher, Les Ulis, France), FITC-conjugated mouse anti-human ILT2 monoclonal antibody (clone GHI/75, Miltenyi, Paris, France), and APC-conjugated mouse anti-human HLA-DR monoclonal antibody (clone REA805, Miltenyi, Paris, France). Non-reactive antibodies of similar species and isotype, coupled with the same fluorochromes, were used as isotypic controls. HLA-DR, PD-L1, CD40, MHC1, HLA-G, ILT2 and ILT4 expression profiles were analyzed using an Astrios cell-sorter (Beckman Coulter, Villepinte, France) or C6 Accuri (BD Biosciences, Le Pont de Claix, France), Attune NxT (Thermo Fisher, Les Ulis, France) or MACSquant (Miltenyi, Paris, France) analyzer. Anti-ILT2 and anti-ILT4 antibodies were used to generate the data presented in Appendix A. Data were analyzed using FlowJo software (BD Biosciences, Le Pont de Claix, France).

### 2.4. Cell Sorting

Adult epidermal keratinocytes were sorted according to HLA-G or PD-L1 expression, using PE-cy7-conjugated mouse anti-human PD-L1 monoclonal antibody (clone MIH1, Thermo Fisher, Les Ulis, France) and Alexa 700-conjugated mouse anti-human HLA-G monoclonal antibody (clone 87G, Novus Biologicals, Lille, France). Appropriate isotype controls were systematically used. Cells were sorted using a FACS Aria 3 sorter (BD Biosciences, Le Pont de Claix, France).

### 2.5. Western Blotting

Total proteins were extracted from cells in lysis buffer: 20 mM Tris, 250 mM NaCl, 2 mM EDTA supplemented with anti-proteases (Roche Molecular Biochemicals, Mannheim, Germany). Cells were lysed by sonication at a concentration of 10^7^ cells/mL for 10 min. Protein concentrations were determined using the Bradford Protein Assay Kit (Bio-Rad, Marnes-la-Coquette, France). In total, 60 µg of protein was separated by electrophoresis in 4–15% Mini Protean TGX Stain free pre-cast gels (Bio-Rad, Marnes-la-Coquette, France). The proteins were transferred onto a nitrocellulose membrane (Bio-Rad, Marnes-la-Coquette, France) using Trans-Blot^®^ SD Semi-Dry Electrophoretic Transfer Cell (Bio-Rad, Marnes-la-Coquette, France). Membranes were blocked by 1 h incubation with TBS Tween 1% and non-fat dry milk 5%. After membrane incubation with primary antibodies for 1 h at room temperature, HRP-conjugated secondary antibodies (Pierce, Thermo Fisher, Les Ulis, France) were used for signal detection in ECL and ECL max (Clarity Western ECL, Bio-Rad, Marnes-la-Coquette, France). The Chemidoc system (Bio-Rad, Marnes-la-Coquette, France) was used for the detection and quantification of Western blot signals. The primary antibodies used were anti-human alpha-tubulin rabbit monoclonal antibody (clone EP1332Y, Abcam, Paris, France), mouse anti-human HLA-G (clone 4H84, Santa Cruz, Heidelberg, Germany), and mouse anti-human HLA-G5 (clone 5A6G7, Thermo Fisher, Les Ulis, France). Membranes were washed 3 times with TBS Tween 0.1% between and after stainings. The 4H84 antibody recognizes all HLA-G isoforms while 5A6G7 recognizes the soluble isoforms HLA-G5 and HLA-G6 [30,31]. The 87G antibody recognizes both the HLA-G1 and HLA-G5 isoforms [32].

### 2.6. Immunofluorescence

For native epidermises extracted from adult donors, 8 µm-thick sections were cut by cryostat. After blocking non-specific antibody binding with BSA 3% for 1 h at room temperature, sections were incubated for 1 h at room temperature with the following primary antibodies: rabbit polyclonal anti-involucrin (IVL) antibody (ab53112, Abcam, Paris, France), and mouse monoclonal anti-laminin 5 (LAM5) (clone P3H9-2, Abcam, Paris, France). Staining was then revealed for 30 min at room temperature using the following secondary antibodies: Alexa Fluor 594 (AF594)-conjugated goat anti-rabbit (Thermo Fisher Scientific, Les Ulis, France), and Alexa Fluor 594 (AF594)-conjugated goat anti-mouse (Thermo Fisher Scientific, Les Ulis, France). Appropriate isotype controls were systematically used. HLA-G1 was stained with Alexa 700-conjugated mouse anti-human HLA-G monoclonal antibody (clone 87G, Novus Biologicals, Lille, France). Nuclei were stained with Dapi (Fluoroshield™, Sigma-Aldrich, Saint-Quentin Fallavier, France). Images were acquired using the AnalysisCellInsight CX7 High-Content Screening (HCS) Platform (Thermo Fisher, Les Ulis, France), or an Axio-observer Z1 (Zeiss, Marly le Roi, France) or SP8 (Leica, Nanterre, France) microscope.

### 2.7. Cell Transduction for Inducible Expression of HLA-G1

Lentiviral transductions were carried out using non-replicative lentiviral vectors (Flash Therapeutics, Toulouse, France). Two vectors were used, for the expression of GFP and doxycycline-inducible RFP (control vector), or of GFP and doxycycline-inducible HLA-G1 (HLA-G1 vector). The maps of the constructions are provided in the Appendix A. Transductions were performed on keratinocytes at 20% confluence, in medium with heat-decomplemented serum. The multiplicity of infection (M.O.I.) was 20, for 24 h at 37 °C, in the presence of hexadimethrine bromide (polybrene) at 8 µg/mL (Sigma-Aldrich, Saint-Quentin Fallavier, France). The cells were then washed 3 times with PBS and amplified in medium for 3 days up to 80% confluence. They were then sorted by flow cytometry (MoFlo ASTRIOS, Beckman Coulter, Villepinte, France) based on GFP expression. The keratinocytes were then amplified to a density of 1000/cm² or deposited in predefined numbers by flow cytometry in 96-well plates.

### 2.8. PBMC Proliferation Measured by 3H-Thymidine Incorporation

Keratinocytes were seeded at various ratios in 96-well culture plates (collagen-1 96-well, BD BioCoat, Le Pont de Claix, France) and incubated for 48 h at 37 °C, in 5% CO_2_ with or without IFN-γ and TNFα. Then, PBMCs were incubated for 1 h at 37 °C, in 5% CO_2_ in 100 μL RPMI medium (Sigma, Saint-Quentin Fallavier, France) supplemented with 20% FCS, enriched in streptomycin and glucose. Then, PBMCs were stimulated or not by anti-CD2:anti-CD3:anti-CD28-coated beads (T-Cell Activation/Expansion Kit, Miltenyi, Paris, France), and seeded or not with keratinocytes, at 100,000 cells per well, with one bead per cell. Sixteen hours before collection on day 5, PBMCs were incubated with tritiated thymidine (20 Ci/mol, PerkinElmer, Boston, MA, USA). Cells were collected on day 5 (FilterMate Harvester, PerkinElmer, Villebon-sur-Yvette, France), and radioactivity was quantified with a β-counter. Keratinocytes’ ability to modulate PBMC proliferation was analyzed by comparing thymidine incorporation by PBMCs in the presence versus absence of keratinocytes.

### 2.9. Flow Cytometry-Based Analysis of CD4^+^ T-Cell Proliferation

Keratinocytes were seeded at various ratios in 96-well culture plates (collagen-1 96-well, BD BioCoat, Le Pont de Claix, France) and incubated for 48 h at 37 °C, in 5% CO_2_ with or without IFN-γ and TNFα. Then, PBMCs were incubated for 1 h at 37 °C, in 5% CO_2_ in 100 μL RPMI medium (Sigma, Saint-Quentin Fallavier, France) supplemented with 20% FCS, enriched in streptomycin and glucose. PBMCs were incubated for 20 min with a proliferation dye (eBioscience Cell Proliferation Dye eFluor 450, Thermo Fisher, Les Ulis, France), then stimulated or not by anti-CD2:anti-CD3:anti-CD28-coated beads (T-Cell Activation/Expansion Kit, Miltenyi, Paris, France), and seeded or not with keratinocytes, at 100,000 cells per well, with one bead per cell. PBMC was quantified by reduction in cell dye intensity after 7 days. Collection was made on day 7, using a flow cytometer. Keratinocytes’ ability to modulate CD4^+^ T-cell proliferation was analyzed by comparing CD4^+^ T-cell dye intensity decrease in the presence versus absence of keratinocytes.

### 2.10. Supernatant Collection

To evaluate the impact of the keratinocyte supernatant on PBMC proliferation, keratinocytes from 3 donors were amplified for 7 days, as specified in the Cell Culture section above. The medium was changed every 2 days. At 80% keratinocyte confluence and after 48 h, the supernatant was collected and centrifuged to remove cellular debris. These supernatants were then used for the PBMC proliferation assays. The same protocol was used with supernatants of fibroblast feeder layers, in order to ascertain the role of each cell type.

### 2.11. Antibody Blocking Experiments

To analyze the role of tolerogenic molecules such as TGFB1, PD-L1 and IL-10 in the inhibitory properties of keratinocytes, cells were incubated at days 1 and 3 with blocking antibodies directed against human TGFB1 (clone 1D11, R&D Systems, Bio-Techne, Noyal Châtillon sur Seiche, France) and PD-L1 (clone 6E11, Genentech, Californie, USA) [30,31,33]. Appropriate isotype controls were systematically used. All CD4^+^ T-cell proliferation assays were performed in triplicate.

### 2.12. Statistics

Significant differences were assessed via 2-tailed Mann–Whitney U-test or *t*-test. All data are presented as mean±SEM. Differences were considered significant for *p* < 0.05; * = *p* < 0.05, ** = *p* < 0.01, *** = *p* < 0.001 and **** = *p* < 0.0001.

## 3. Results

### 3.1. Amplified Keratinocytes Are Hypoimmunogenic and Limit CD4^+^ T-Cell Proliferation

The expression of molecules that co-stimulate T-cells (i.e., HLA-I, HLA-DR, CD86, and CD40) was first analyzed by flow cytometry in amplified keratinocytes. Keratinocytes expressed HLA-I and CD40, but displayed very low levels of HLA-DR and CD86 (Figure 1A). Keratinocytes therefore lack critical T-cell stimulation molecules when cultured under standard amplification conditions.

We then analyzed both the immunogenic and immunosuppressive properties of amplified keratinocytes under allogeneic conditions, i.e., facing HLA-mismatched PBMCs. Keratinocytes were first incubated with PBMC for 5 days to evaluate their ability to induce PBMC alloproliferation. PBMCs activated by CD3^+^ and CD28^+^ beads were used as positive controls for PBMC proliferation. At the highest quantity (i.e., 25,000 cells), keratinocytes were not able to induce PBMC alloproliferation (*p* < 0.05, *n* = 3 distinct keratinocyte donors) (Figure 1B). Secondly, to assess the immunomodulatory properties of keratinocytes, we added them to bead-activated PBMCs. The results show lower PBMC proliferation in the presence of keratinocytes (Figure 1B). Then, to target the inhibition of CD4^+^ T-cell proliferation by keratinocytes, activated PBMCs pre-stained with a proliferation dye marker were incubated with serial ratios of keratinocytes. Keratinocytes ratio-dependently limited CD4^+^ T-cell proliferation (*p* < 0.05, *n* = 3 PBMC donors) (Figure 1C,D). Thus, amplified keratinocytes appear hypoimmunogenic in vitro and display immunomodulatory properties, such as the inhibition of CD4^+^ T-cell proliferation.

### 3.2. Amplified Keratinocytes Limit CD4^+^ T-Cell Proliferation Whether in Inflammatory Conditions or Not

We then investigated whether the immune properties of keratinocytes varied according to the inflammatory context. To mimic inflammatory conditions, keratinocytes from one representative donor were amplified in the presence of 10 ng/mL IFNγ and TNF-α pro-inflammatory cytokines. To determine the biological effects of the two cytokines, we measured HLA-DR expression at the keratinocyte cell surface by flow cytometry (mean + SEM, *p* < 0.05, *n* = 4). As expected, we observed HLA-DR overexpression in keratinocytes stimulated by IFNγ and TNF-α (Figure 2A). In order to evaluate the impact of IFNγ and TNF-α on their immune properties, keratinocytes previously stimulated with 10 ng/mL IFNγ and TNF-α or not stimulated were incubated with activated PBMCs. The amplified keratinocytes retained their ability to inhibit PBMC proliferation in the presence of IFNγ and TNF-α (mean + SEM, *p* < 0.05, *n* = 3) (Figure 2B). We then measured the inhibition of CD4^+^ T-cell proliferation by various quantities of keratinocytes, stimulated by 10 ng/Ml IFNγ and TNF-α or not. There was no difference in CD4^+^ T-cell proliferation inhibition between keratinocytes with or without IFNγ and TNF-α (mean + SEM, *p* < 0.05, *n* = 3) (Figure 2C,D). Amplified keratinocytes therefore limit CD4^+^ T-cell proliferation independently of inflammatory conditions.

### 3.3. Immunomodulation Mediated by Amplified Keratinocytes Involves Soluble Factors

Next, we sought to identify the mechanisms involved in immunomodulation mediated by keratinocytes. To study the role of soluble factors, keratinocytes were amplified and the culture supernatant was collected. The results show that PBMC proliferation was inhibited in the presence of the keratinocyte supernatant (mean + SEM, *p* < 0.05, *n* = 3) (Figure 3A), showing that the immunomodulatory properties of amplified keratinocytes involve soluble factors. Then, activated PBMCs pre-stained with a proliferation dye marker were incubated with increasing dilutions of keratinocyte supernatant. The supernatants dose-dependently limited CD4^+^ T-cell proliferation (mean + SEM, *p* < 0.05, *n* = 3) (Figure 3B,C). As keratinocytes were amplified with a feeder layer of fibroblasts, we tested whether the supernatant from the fibroblasts could be involved in this inhibition. Experiments using the fibroblast supernatant did not show any immunomodulation of PBMC proliferation (see Appendix A). The immunomodulating effect was therefore due to soluble factors produced by the keratinocytes. It has previously been shown that TGFB1 is a soluble factor that reduces mouse T-cell proliferation [32] and which is known to be produced by human epidermal keratinocytes [34]. In this regard, we were able to detect TGFB1 expression by amplified keratinocytes (Figure 3E). To assess the role of this cytokine in the inhibitory effect observed with the keratinocyte supernatant, we performed blocking using the anti-TGFB1 antibody. Anti-TGFB1 partially reversed the inhibition of CD4^+^ T-cell proliferation in the presence of the keratinocyte supernatant (Figure 3F). Thus, the immunosuppressive properties of amplified keratinocytes involve soluble factors, including TGFB1.

### 3.4. Immunomodulation by Amplified Keratinocytes Involves the PD-L1 Immune Checkpoint Protein

We then considered whether the immunosuppressive properties of keratinocytes also involved HLA-G and PD-L1. Since HLA-G and PD-L1 are well-described immune checkpoint molecules involved in T-cell proliferation inhibition, we looked for their expression by amplified keratinocytes. Flow cytometry analysis showed that both PD-L1 and HLA-G were expressed at the keratinocyte cell surface (Figure 4A). While PD-L1 was strongly expressed by amplified keratinocytes, HLA-G expression was much lower, in terms of both intensity and number of positive cells (Figure 4B). We then investigated which HLA-G isoforms were expressed by keratinocytes. Western blot analysis using the 4H84 anti-HLA-G antibody showed a band corresponding to the membrane-bound HLA-G1 isoform (39 KDa) (see Appendix A). To verify the lack of expression of the soluble HLA-G5 isoform, we used the 5A6G5 antibody, which selectively recognizes this isoform. No band was detected in keratinocyte lysates. Taken together, these data showed that keratinocytes express the membrane-bound HLA-G1 isoform. In other cellular models, such as mesenchymal stem cells, IFNγ and TNF-α, licensing increased the expression of immune checkpoints such as HLA-G [35]. Similarly, we demonstrated the overexpression of both HLA-G1 and PD-L1 molecules on keratinocytes treated with IFNγ and TNF-α (mean + SEM, *p* < 0.05, *n* = 4) (Figure 4C).

To evaluate the respective roles of HLA-G1 and PD-L1 in keratinocyte-mediated CD4^+^ T-cell proliferation inhibition, we sorted cells according to HLA-G1 or PD-L1 expression. Activated PBMCs pre-stained with a proliferation marker were incubated with the same numbers of HLA-G^−^ PD-L1^−^, HLA-G^+^ PD-L1^−^, HLA-G^−^ PD-L1^+^ or HLA-G^+^ PD-L1^+^ keratinocytes (Figure 4D). PD-L1^+^ keratinocytes inhibited CD4^+^ T-cell proliferation (Figure 4E), and inhibition was slightly greater when keratinocytes expressed both checkpoints. Notably, blocking PD-L1 restored CD4^+^ T-cell proliferation (Figure 4F). Unfortunately, since no reliable HLA-G-blocking antibody is commercially available, we could not formally confirm the role of this immune checkpoint. Thus, the immunosuppressive properties of amplified keratinocytes involve at least the PD-L1 immune checkpoint molecule.

### 3.5. Tissue keratinocytes Express HLA-G1, Which Contributes to Their Immunomodulatory Properties

We investigated whether the immunosuppressive properties observed in amplified keratinocytes were biologically relevant, in terms of keratinocytes directly extracted from the skin tissue. Cells from nine healthy donors were extracted from skin tissue but were not amplified ex vivo. Interestingly, there was no expression of PD-L1 on tissue keratinocytes, whereas HLA-G1 expression was detected by flow cytometry (mean + SEM, *n* = 9) (Figure 5A,B). Notably, when EGF was removed from the keratinocyte amplification medium, PD-L1 expression decreased sharply, demonstrating that at least EGF contributes to the expression of PD-L1 in amplified keratinocytes (see Appendix A). In terms of immunohistochemistry, HLA-G1 was found to be expressed throughout the living layers of the epidermis (Figure 5C). HLA-G1 expression was localized between that of laminin 5, a basal lamina marker, and that of involucrin, a horny-layer marker. We then assessed the contribution of HLA-G1 to tissue keratinocyte-driven immunomodulation. Tissue-extracted keratinocytes were sorted according to HLA-G1 expression and tested for activated allogeneic PBMCs. HLA-G^+^ tissue keratinocytes inhibited CD4^+^ T-cell proliferation more strongly than HLA-G^−^ keratinocytes (mean + SEM, *p* < 0.05, *n* = 3) (Figure 5D,E). Notably, there was no overexpression of PD-L1 and TGFB1 in keratinocytes following the addition of doxycycline (see Appendix A). The immune checkpoint HLA-G1 thus contributes to the inhibition of CD4^+^ T-cell proliferation by tissue keratinocytes.

### 3.6. Inducible Expression of HLA-G1 in Amplified Keratinocytes Increases Their Immunomodulatory Properties

Since the HLA-G1 expression by tissue and amplified keratinocytes is low, we wondered whether the associated immunomodulation could be increased by boosting HLA-G1 expression. We therefore designed a lentiviral vector to induce HLA-G1 expression after the addition of doxycycline and compared it to a control vector that inducibly expressed RFP (Figure 6A). Transduced cells from one representative skin donor were amplified for 7 days. Flow cytometry showed strong inducible HLA-G1 expression in keratinocytes after the addition of doxycycline (mean + SEM, *n* = 3) (Figure 6B). This was confirmed by Western blot analysis using the 4H84 mAb, in which monomers as well as dimers (functional proteins) of HLA-G1 were detected. Detection of α-tubulin was used as the positive control (mean + SEM, *p* < 0.0001, *n* = 3) (Figure 6C). K562 cells transduced with HLA-G1 were used as HLA-G^+^-positive controls. No toxicity of doxycycline on keratinocytes was observed (data not shown). We therefore generated keratinocytes that could be induced to express a high level of cell-surface HLA-G1.

Finally, we examined whether transduced keratinocytes displaying inducible HLA-G1 expression were more immunosuppressive than regular amplified keratinocytes. The results showed that the transduced keratinocytes induced significantly greater inhibition of CD4^+^ T-cell proliferation after doxycycline treatment than transduced keratinocytes in the absence of doxycycline (mean + SEM, *p* < 0.05, *n* = 3) (Figure 6D,E). There was no difference between non-transduced keratinocytes and keratinocytes that were transduced but not induced. Thus, overexpressing HLA-G1 in keratinocytes increased their immunosuppressive properties.

## 4. Discussion

We here present results showing that human skin keratinocytes are able to inhibit CD4^+^ T-cell proliferation via secretion of TGFB1 as well as expression of HLA-G1 and PD-L1 immune checkpoints. It is notably the first time that the expression of HLA-G protein has been described in healthy human epidermis.

The keratinocytes obtained from skin samples from healthy donors expressed immune markers on their surface. Over 95% expressed MHCI; in contrast, less than 5% expressed HLA-DR or CD86, and a few expressed the CD40 costimulatory molecule. The capacity of keratinocytes to perform antigen presentation to CD4^+^ T-cells through HLA class II expression in vitro is therefore limited. On the other hand, keratinocytes showed immunosuppressive properties in vitro, reducing CD4^+^ T-cell proliferation in allogeneic conditions. Although differences in the frequency of divided CD4^+^ T-cells varied according to each PBMC donor, inhibition mediated by keratinocytes was observed in all cases. Adding IFN-γ and TNF-α to mimic an inflammatory context altered keratinocyte HLA marker expression. Keratinocytes notably overexpress HLA-DR, as previously shown with mesenchymal stem cells (MSCs) [35]. However, even under these inflammatory conditions, keratinocyte-mediated CD4^+^ T-cell proliferation inhibition was still observed.

The inhibition of CD4^+^ T-cell proliferation by keratinocytes appeared to be mediated by soluble factors, such as TGFB1, and by immune checkpoints, including PD-L1 and HLA-G1. Interestingly, HLA-G and TGFB1 have been shown to be interdependent, as HLA-G upregulates TGFB1 expression and vice versa, constituting an auto-induction loop that amplifies the tolerogenic background. [36,37]. TGFB1 reduces T-cell activation and proliferation through the Smad3 pathway [37,38,39]. Thus, the TGFB1 produced by keratinocytes probably contributes to their ability to inhibit CD4^+^ T-cell proliferation. These TGFB immunosuppressive properties were also described for bronchial epithelial cells, in combination with IL-10 and HLA-G [33].

Studies in mice showed that skin keratinocytes expressing PD-L1 reduced T-cell proliferation and effector function at local inflammatory sites [40]. PD-L1 expression also correlates with a higher rate of Treg cells in the skin [41]. PD-L1 binds to the PD-1 expressed on the surface of T-cells, which in turn inhibits their activity [11] and limits autoimmune reactions [12]. Here, we demonstrated the strong expression of PD-L1 in amplified keratinocytes, contributing to their ability to modulate CD4^+^ T-cell proliferation. However, we did not observe any PD-L1 expression in keratinocytes directly extracted from the tissue without in vitro amplification. The induction of PD-L1 expression by keratinocytes during amplification in culture can be explained by the presence of growth factors such as EGF, as demonstrated here and also shown in other systems [42,43]. However, keratinocytes may be less exposed to EGF once they are grafted onto the patient, decreasing their expression of PD-L1 and its associated tolerogenic functions, and thus increasing the risk of rejection.

The epidermal expression of HLA-G was previously described in pathological contexts, such as skin cancer or inflammatory dermatosis [38]. HLA-G1 is also expressed in healthy human skin at the transcript level [39], but demonstration at the protein level has been lacking. HLA-G was detected in psoriasis, mainly in the macrophages of the dermo-epidermal junction [44]. HLA-G expression was also studied in patients with atopic dermatitis [45]. In this case, HLA-G was expressed by T-cells infiltrating tissue, macrophages and dendritic cells, and this expression of HLA-G could be favored by the expression of IL-10. In pemphigus vulgaris, a rare autoimmune skin disease, deletion in the HLA-G gene was found to be elevated at exon 8 [46,47]. There were also significantly higher soluble HLA-G levels in the circulating blood of patients with systemic lupus erythematosus [48]. HLA-G was also found in skin cancers, such as cutaneous lymphoma and skin basal and squamous cell carcinoma [49,50,51]. This expression has been particularly studied in melanoma—melanoma cells have less MHCI, which could make them more susceptible to lysis by NK cells. However, HLA-G expressed by melanoma cells was shown to protect them from NK lysis [52,53,54,55,56].

The present study demonstrated for the first time the expression of HLA-G1 protein by keratinocytes in the epidermis of several healthy donors. This successful detection may have been due to the use of more sensitive methods than in previous studies, and notably fluorescence-based assays. No expression of HLA-G receptors ILT2 or ILT4 was observed on amplified keratinocytes (see Appendix A), which supports the following mechanism: the action of HLA-G1 expressed in keratinocytes is not involved in an autocrine loop, but rather consists of the modulation of immune cell populations.

Functionally, we showed that keratinocytes from skin tissue reduced CD4^+^ T-cell proliferation through HLA-G1 expression. However, fewer than 10% of keratinocytes expressed HLA-G1 in the tissue. With a view to being able to use keratinocytes for skin allografts, we tried overexpressing this immune checkpoint in order to increase the immunosuppressive properties of amplified keratinocytes. Here, we showed that HLA-G1 expression obtained with an inducible vector boosted the immunosuppressive properties of amplified keratinocytes. These immunosuppressive cells used in allogeneic conditions could open up new therapeutic prospects for patients with severe burns or with a strong inflammatory context, in whom autologous skin transplant is not feasible. Thus, the transient use of amplified keratinocytes strongly overexpressing HLA-G could make a stop-gap allogeneic graft immediately available, in order to reduce inflammation and leave time for an autologous graft to be prepared. As amplified keratinocytes also strongly express PD-L1, this may boost the immunosuppressive effects of HLA-G, in cooperation with TGFB1. Indeed, in the context of tumors, we recently reported that PD-L1 and HLA-G target distinct T-cell subpopulations. [57,58].

Taken together, the present results showed that human skin keratinocytes inhibit the proliferation of CD4^+^ T-cells via the expression of HLA-G1 and PD-L1 immune checkpoints. Increasing the expression of HLA-G1 further enhances immunomodulation, opening perspectives for the bioengineering of temporary allogeneic skin grafts. In current clinical strategies, grafted tissues that can contain autologous or allogenic cellular components are not engineered to contribute to reducing the immune response when alone. At present, pharmacological treatments are available to keep acute and chronic immune reactions, including inflammation, under control. This specific research contribution offers new perspectives for next-generation cell therapy approaches, aiming at improving the control of tolerogenicity in skin substitute grafting, via the design of HLA–G1–vehicle cells. The prospects concern skin grafting in severe burns and irradiated patients, as well as more general applications in reconstructive surgery, including chronic wounds, and breast and maxillo-facial reconstruction.

## Figures and Tables

**Figure 1 cells-10-01438-f001:**
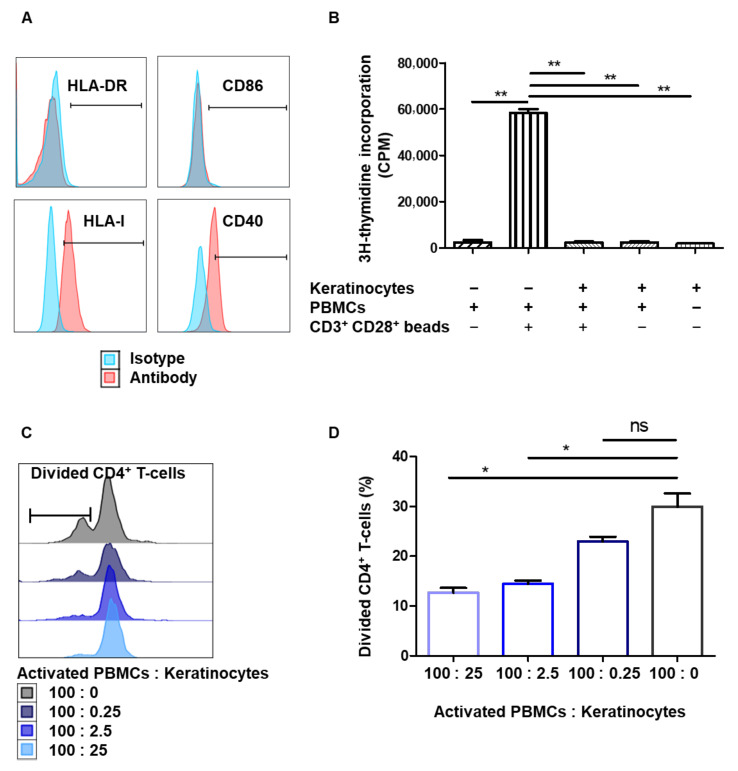
Amplified keratinocytes are hypoimmunogenic and limit CD4^+^ T-cell proliferation. (**A**) The keratinocytes were amplified for 7 days in a serum+feeder medium. Representative flow cytometry profiles of HLA-DR, HLA-I, CD40 and CD86 are shown. (**B**) In total, 25,000 keratinocytes from 3 different donors were incubated or not with 100,000 PBMCs for 5 days. PBMCs activated by CD3^+^ CD28^+^ beads were used as positive controls. PBMC proliferation was quantified by thymidine incorporation at day 5 (mean + SEM, *p* < 0.05, *n* = 3). (**C**,**D**). In total, 25,000, 2500, 250 or 0 keratinocytes of one representative donor were incubated with 100,000 PBMCs for 7 days. PBMCs were pre-marked with a proliferation dye, and activated by CD3^+^ CD28^+^ beads. PBMC proliferation was quantified by dye reduction at day 7. (**C**) Representative flow cytometry profiles at day 7. (**D**) CD4+ T-cell proliferation was inhibited dose-dependently by keratinocytes (mean + SEM, *p* < 0.05, *n* = 3 PBMC). Exact *p*-values were determined using the *t*-test. *: *p* < 0.05 and **: *p* < 0.01.

**Figure 2 cells-10-01438-f002:**
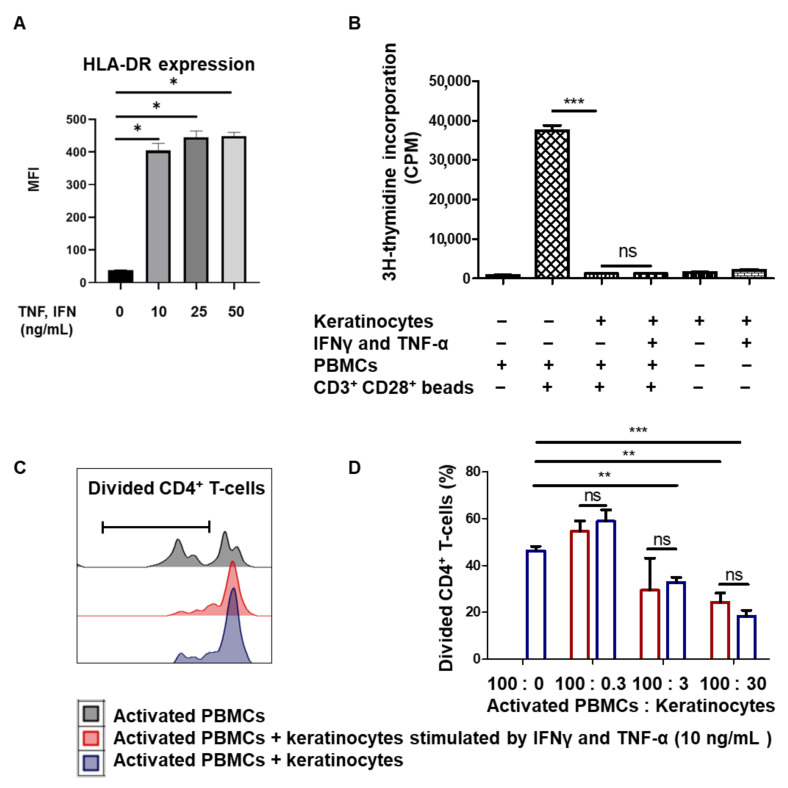
Amplified keratinocytes limit CD4^+^ T-cell proliferation whether they are in inflammatory conditions or not. (**A**) Cells from one representative donor were cultured for 2 days with IFNγ and TNF-α. HLA-DR expression was analyzed by flow cytometry (mean + SEM, *p* < 0.05, *n* = 4; Mann–Whitney U-test). (**B**) In total, 20,000 keratinocytes from 1 representative donor, stimulated or not by 10 ng/mL IFNγ and TNF-α for 2 days, were incubated or not with 100,000 PBMCs for 5 days. PBMCs activated by CD3^+^ CD28^+^ beads were used as positive controls. PBMC proliferation was quantified by thymidine incorporation at day 5 (mean + SEM, *p* < 0.05, *n* = 3). (**C**,**D**) Keratinocytes stimulated or not by 10 ng/mL IFNγ and TNF-α for 2 days were incubated with 100,000 PBMCs for 7 days. PBMCs were activated by CD3^+^ CD28^+^ beads. PBMC proliferation was quantified by dye decrease at day 7. (**C**) Representative flow cytometry profiles at day 7. (**D**) CD4^+^ T-cell proliferation according to inflammatory conditions (mean + SEM, *p* < 0.05, *n* = 3). Exact p-values were determined using the *t*-test. *: *p* < 0.05 and **: *p* < 0.01 and ***: *p* < 0.001.

**Figure 3 cells-10-01438-f003:**
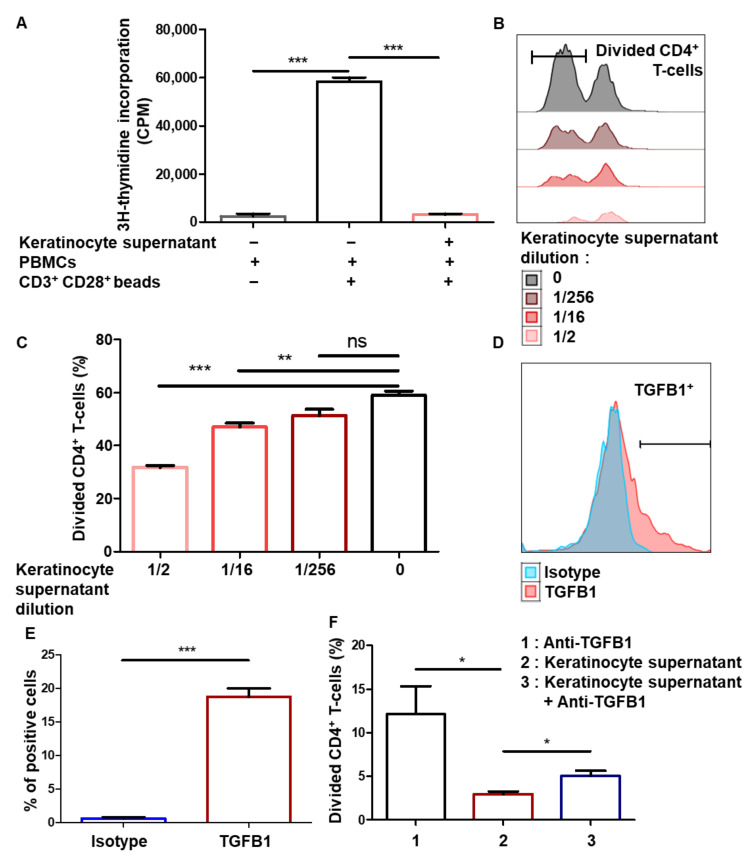
Immunomodulation mediated by amplified keratinocytes involves soluble factors. (**A**) Keratinocytes from 3 different donors were amplified for 2 days. Then, supernatants were incubated or not with 100,000 PBMCs for 5 days. PBMCs activated by CD3^+^ CD28^+^ beads were used as positive controls. PBMC proliferation was quantified by thymidine incorporation at day 5 (mean + SEM, *p* < 0.05, *n* = 3). (**B,C**) Different concentrations of keratinocyte supernatant from 1 representative donor were incubated with 100,000 PBMCs for 7 days. PBMCs were pre-marked with a proliferation dye. PBMCs were activated by CD3^+^ CD28^+^ beads. PBMC proliferation was quantified by dye decrease at day 7. (**B**) Representative flow cytometry profiles at day 7. (**C**) CD4+ T-cell proliferation inhibition as a function of keratinocyte supernatant dilution (mean + SEM, *p* < 0.05, *n* = 3). (**D**,**E**) Flow cytometry analysis of TGFB1 levels in keratinocytes. (**F**) CD4^+^ T-cell proliferation assay with TGFB1 blocking. Exact p-values were determined using the *t*-test. *: *p* < 0.05 and **: *p* < 0.01 and ***: *p* < 0.001 *n* = 3.

**Figure 4 cells-10-01438-f004:**
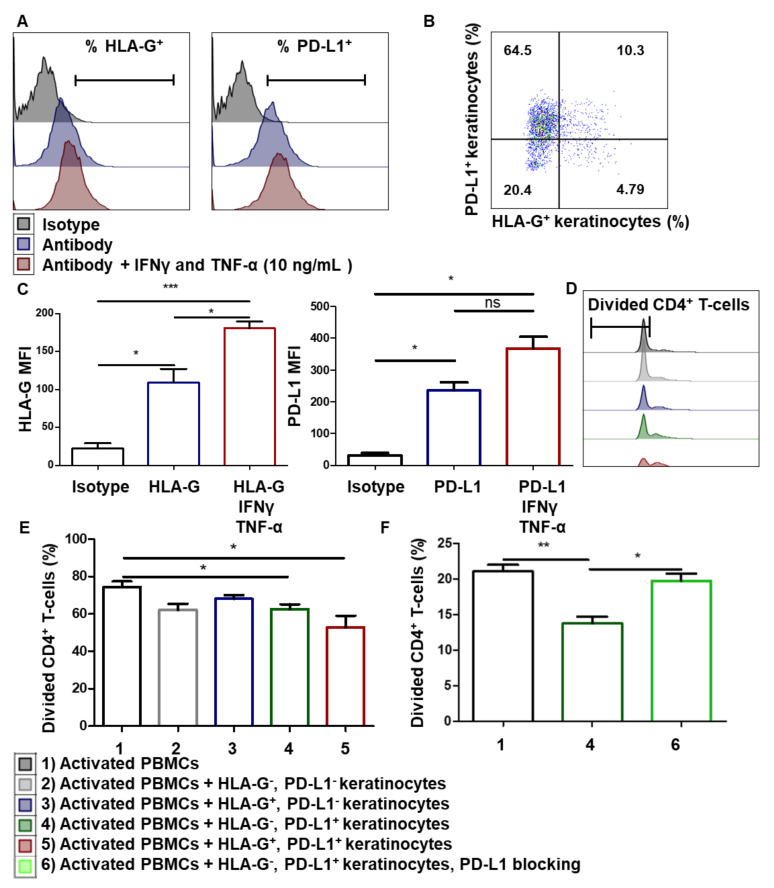
Immunomodulation by amplified keratinocytes involves the PD-L1 immune checkpoint. (**A**–**C**) Cells from 3 donors were cultured for 2 days with IFNγ and TNF-α. (**A**) Representative flow cytometry profiles at day 2. (**B**) Expression of PD-L1 and HLA-G on keratinocytes. (**C**) Analysis of HLA-G1 (87G) and PD-L1 expression was performed by flow cytometry (mean + SEM, *p* < 0.05, *n* = 3). Paired *t*-test. *: *p* < 0.05 and **: *p* < 0.001. (**D**–**F**) In total, 10,000 keratinocytes sorted as HLA-G^−^, HLA-G^+^, PD-L1^−^ or PD-L1^+^ were incubated with 100,000 PBMCs for 7 days. PBMCs were activated by CD3^+^ CD28^+^ beads. PBMC proliferation was quantified by dye decrease at day 7. (**D**) Representative flow cytometry profiles at day 7. (**E**) CD4^+^ T-cell proliferation inhibition varied according to the presence of HLA-G or PD-L1 (mean + SEM, *p* < 0.05, *n* = 3). (**F**) Effect of PD-L1 blocking. Exact p-values were determined using the *t*-test. *: *p* < 0.05 and **: *p* < 0.01 and ***: *p* < 0.001, *n* = 3.

**Figure 5 cells-10-01438-f005:**
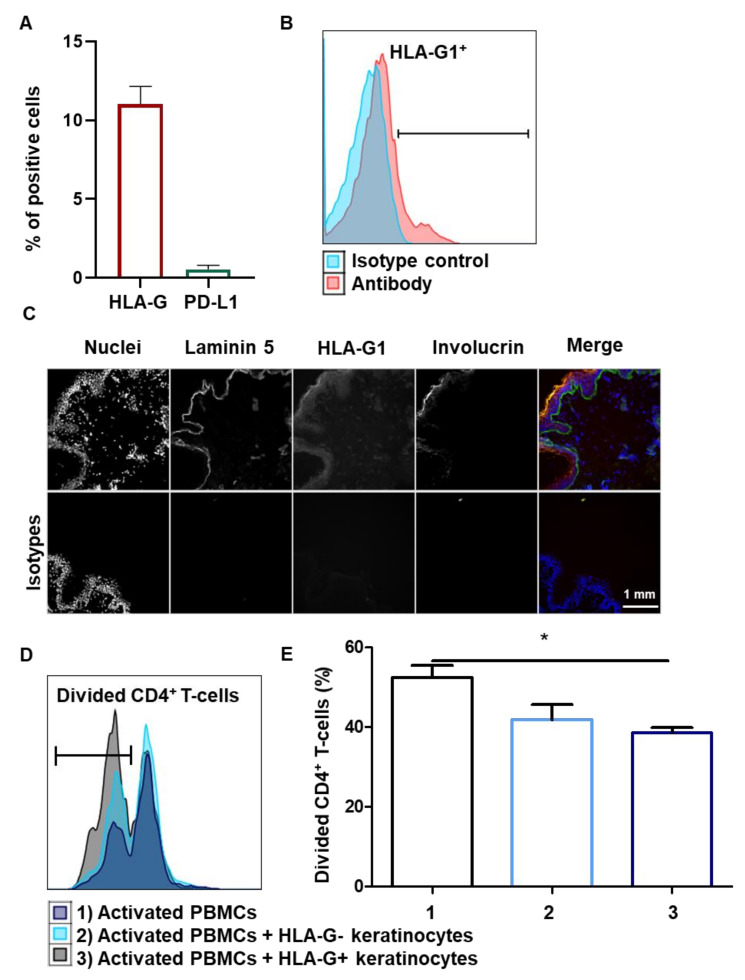
Tissue keratinocytes express HLA-G1, making them immunomodulators. (**A**,**B**) Cells from 9 representative adult donors were extracted. (**A**) Analysis by flow cytometry of PD-L1 and HLA-G1 expression on keratinocytes (mean + SEM, *n* = 9). Exact p-values were determined via Wilcoxon matched-pairs signed rank test. (**B**) Representative profiles of HLA-G1 (87G) expression on keratinocytes just extracted from tissue. (**C**) Expression of HLA-G in the whole epidermis (immunohistochemistry). Laminin 5 is a basal lamina marker and involucrin is a horny-layer marker. (**D**,**E**) In total, 10,000 keratinocytes just extracted from tissue, sorted as HLA-G^−^ or HLA-G^+^, were incubated with 100,000 PBMCs for 7 days. PBMCs were activated by CD3^+^ CD28^+^ beads. PBMC proliferation was quantified by dye decrease at day 7. (**D**) Representative flow cytometry profiles at day 7. (**E**) CD4^+^ T-cell proliferation inhibition varied according to presence of keratinocytes and HLA-G (mean + SEM, *p* < 0.05, *n* = 3). Exact p-values were determined using the *t*-test. *: *p* < 0.05.

**Figure 6 cells-10-01438-f006:**
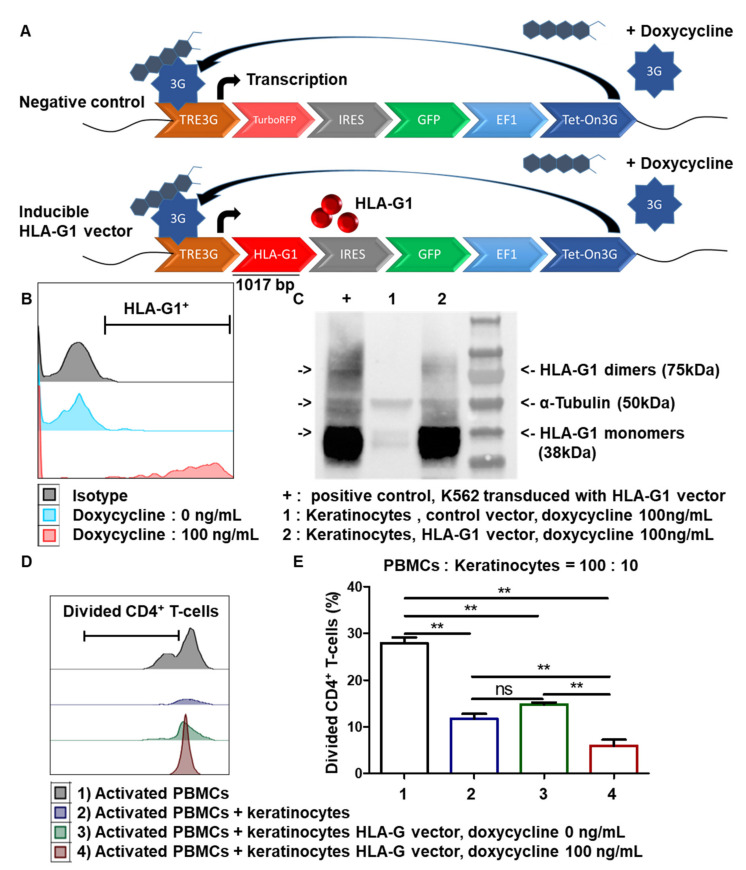
Inducible expression of HLA-G1 increases immunomodulation by amplifying keratinocytes. (**A**) Construction of vectors allowing inducible expression of HLA-G1 by keratinocytes. (**B**,**C**) Transduced cells from 1 representative donor were cultivated for 7 days after transduction. HLA-G1 expression on keratinocytes was induced by doxycycline. (**B**) Representative profiles of HLA-G1 expression on transduced cells. (**C**) Typical gel photograph corresponding to different cultures, with α-tubulin detection as loading control. K562 cells transduced with HLA-G1 were used as positive control with 4H84 antibody. (**D**,**E**) In total, 10,000 keratinocytes, transduced or not with HLA-G vector, were incubated with 100,000 PBMCs for 7 days. PBMCs were pre-marked with a proliferation dye. PBMCs were activated by CD3^+^ CD28^+^ beads. PBMC proliferation was quantified by dye decrease at day 7. (**D**) Representative flow cytometry profiles at day 7. (**E**) CD4+ T-cell proliferation depending on presence of keratinocytes and HLA-G1 (mean + SEM, *p* < 0.05, *n* = 3). Exact p-values were determined using the *t*-test **: *p* < 0.01, *n* = 3.

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
