# Peer review of "Human Keratinocytes Inhibit CD4+ T-Cell Proliferation through TGFB1 Secretion and Surface Expression of HLA-G1 and PD-L1 Immune Checkpoints"

_cells, 2021, doi:10.3390/cells10061438_

Round 1

Reviewer 1 Report

This paper analyzes the immunomodulating activity of human skin keratinocytes on CD4+ T-cells.

In this study, human skin tissue was collected from healthy donors after breast reduction surgery, and keratinocytes were used directly (tissue keratinocytes) or after amplification in culture (amplified keratinocytes).

Cells were analyzed via flow cytometry, sorted according to surface-expressed markers. Western blot signals were quantified, and immunofluorescence images were collected. Keratinocyte modulation on CD4+ T-cell proliferation was assessed via flow cytometry of CD4+ T-cells seeded with or without keratinocytes.

Thymidine incorporation was measured to compare PBMC proliferation in presence or absence of keratinocytes.

Results showed a dose-dependent ability of keratinocytes to lower CD4+ T-cell proliferation via secretion of TGFB1 and expression of HLA-G1 (and PDL1 immune checkpoints. This was also found when keratinocytes were amplified in presence of IFN to mimic inflammatory conditions, and this spreads the clinical applicability of this new finding. PBMC proliferation was also found to be lower in presence of keratinocytes.

The bibliography is complete and appropriate, although a more specific analysis of the clinical applicability of the results could improve the scientific contribution of this work and the interest of the reader. The English used is correct and readable.

In conclusion, this work is well conducted, and it brings a significant contribution to the field. In fact, it lays the foundations for clinical studies on the use of transient bioengineered allogenic grafts to reduce inflammation in early treatment of patients with severe burns.

Author Response

Bold black font: reviewer’s comments

Black font: author’s responses

Red font: text additions in the manuscript

Reviewer 1

In conclusion, this work is well conducted, and it brings a significant contribution to the field. In fact, it lays the foundations for clinical studies on the use of transient bioengineered allogenic grafts to reduce inflammation in early treatment of patients with severe burns.

We thank reviewer 1 for his favorable appreciation of our work.

A more specific analysis of the clinical applicability of the results could improve the scientific contribution of this work and the interest of the reader.

We have considered this comment and added the following concise paragraph at the end of the ‘Discussion‘:

In current clinical strategies, grafted tissues that can contain autologous or allogenic cellular components are not engineered to contribute on their own to reducing the immune response. At present, pharmacological treatments are available to maintain acute and chronic immune reactions, including inflammation, under control. This specific research contribution opens perspectives for next-generation cell therapy approaches, aiming at improving the control of tolerogenicity in skin substitute grafting, via the design of HLA-G1-vehicle cells. The prospects concern skin grafting in severe burns and irradiated patients, as well as more general applications in reconstructive surgery, including chronic wounds, and breast and maxillo-facial reconstruction.

Reviewer 2 Report

The manuscript entitled “Human keratinocytes inhibit CD4+ T-cell proliferation through TGFB1 secretion and surface expression of HLA-G1 and PD-L1 immune checkpoints” describe finding that HLA-G1 is expressing in healthy skin epidermis and contribute to the immunosuppression. The authors also found TGFB1 secretion and PD-L1 expression is independently important for CD-4+ T-cell proliferation. Furthermore, they tried overexpression of HLA-G1 in keratinocytes and proposed application for skin transplantation under immunosuppressive condition. The experimental motive and concept are challenging and interesting, but English and literature is not well written. The manuscript needs careful editing.   

Major points

Definition of divided CD4+ T-cells in figure 4E looks difference with the other figures as figure 1C, 2C, 3B, 5D, and 6D. Please explain this difference. Moreover, please explain about differences of frequency of divided CD4+ T-cells in each experiment.

Minor points

  1. The authors used term “HLA-G” and “HLA-G1”. Please define and describe the difference in Introduction section. (Dose HLA-G antibody recognize all isoforms such as HLA-G1 and HLA-G5?)

  1. In materials and methods, 2.6. Immunofluorescence section mentioned about anti-filaggrin antibody, but no data about it. In same section, please describe correct name of Laminin 5. There are some mistakes of describing. e.g., ILT4 and ILT5 antibodies are described in materials and methods but no data in main text.

  1. Treated condition of figure 4D such as 1) Activated PBMCs….. is little far. Please integrate D, E, and F to the near place.

  1. Please indicate expression level of PD-L1 and HLA-G1 in 9 donors. If you do not have data, please explain it.

  1. Please check expression level of PD-L1 and TFGB after dox treatment by WB or flowcytometry.

Author Response

Bold black font: reviewer’s comments

Black font: author’s responses

Red font: text additions in the manuscript

Reviewer 2

The manuscript needs careful editing

We apologize for that, the revised version has been carefully checked by a nature English-speaking scientist corrector.

Major point

Definition of divided CD4+ T-cells in figure 4E looks difference with the other figures as figure 1C, 2C, 3B, 5D, and 6D. Please explain this difference. Moreover, please explain about differences of frequency of divided CD4+ T-cells in each experiment.

We thank the reviewer for his remark. The differences in the frequency of divided CD4 + T cells vary with each PBMC donor. However, keratinocyte-mediated inhibition is observed in all cases. It should be noted that experiments were carried out on 3 different PBMC donors. To clarify this point, the above comment has been added to the revised version of the manuscript.

This comment has been added in the ‘Discussion’:

Although differences in frequency of divided CD4+ T-cells varied according to each PBMC donor, inhibition mediated by keratinocytes was observed in all cases.

Minor points

  1. The authors used term “HLA-G” and “HLA-G1”. Please define and describe the difference in Introduction section. (Dose HLA-G antibody recognize all isoforms such as HLA-G1 and HLA-G5?)

The HLA-G primary transcript is alternatively spliced giving rise to 7 spliced transcripts which lead to 7 protein isoforms. Four of them are membrane-bound (HLA-G1 to HLA-G4), and 3 are soluble (HLA-G5 to HLA-G7). The HLA-G1 protein corresponds to the full-length isoform having three extracellular domains (α1, α2, and α3). It is also the main HLA-G isoform expressed in vivo. We have used in the text “HLA-G” as a general name to present HLA-G, and “HLA-G1 or HLA-G5” to specify which isoform is analyzed. [26]. Regarding anti-HLA-G antibodies used in our experiments, the 4H84 mAb recognizes all HLA-G isoforms while the 5A6G7 mAb recognizes specifically the HLA-G5 and -G6 soluble isoforms [31,32]. 87G recognizes the HLA-G1 and G5 isoforms [33]. These details have been provided in the revised version of the manuscript.

As requested, we have introduced these points in the manuscript.

In the ‘Introduction’:

The HLA-G primary transcript is alternately spliced, giving rise to 7 spliced transcripts which lead to 7 protein isoforms. The HLA-G1 to G4 isoforms are membrane-bound, while the HLA-G5 to G7 isoforms are their soluble counterparts. The HLA-G1 protein is the full-length isoform, having three extracellular domains (α1, α2, and α3), and it is also the main HLA-G isoform expressed in vivo [26].

In the ‘Materials and Methods’:

The 4H84 antibody recognizes all HLA-G isoforms while 5A6G7 recognizes the soluble isoforms HLA-G5 and HLA-G6 [31,32]. The 87G antibody recognizes both the HLA-G1 and HLA-G5 isoforms [33].

  1. In materials and methods, 2.6. Immunofluorescence section mentioned about anti-filaggrin antibody, but no data about it. In same section, please describe correct name of Laminin 5. There are some mistakes of describing. e.g., ILT4 and ILT5 antibodies are described in materials and methods but no data in main text.

We apologize for these mistakes and lack of clarity, which has been corrected in the ‘Materials and Methods’:

anti-fillaggrin’ has been replaced by ‘mouse monoclonal anti-laminin 5 (LAM5) (clone P3H9-2, Abcam)’

Concerning the ILT2 and ILT4, these antibodies were used to generate the data presented in a new supplementary figure (Figure S6).

Accordingly, the following sentence has been added in ‘Materials and Methods’:

Anti-ILT2 and anti-ILT4 antibodies were used to generate the data presented in Figure S6.

The new Figure S6 is also mentioned in the discussion:

‘ … No expression of HLA-G receptors ILT2 or ILT4 was observed on amplified keratinocytes (see Supplementary Materials Figure S6), which supports the following mechanism: the action of HLA-G1 expressed in keratinocytes is not involved in an autocrine loop but rather consists in modulation of immune cell populations. …’;

  1. Treated condition of figure 4D such as 1) Activated PBMCs….. is little far. Please integrate D, E, and F to the near place.

We thank the reviewer for his comment and have corrected the Figure 4.

  1. Please indicate expression level of PD-L1 and HLA-G1 in 9 donors. If you do not have data, please explain it.

The data presented in Figure 5A correspond to Means +/- SD calculated from experiments performed on 9 donors.

  1. Please check expression level of PD-L1 and TGFB after dox treatment by WB or flow cytometry.

We thank the reviewer for his comment and have carried out additional experiments to answer this point. Data are now presented in Supplementary Materials Figure S5). Results showed no overexpression of PD-L1 and TGFB1 in keratinocytes following doxycycline addition. 

The following sentence has been added in the ‘Results’ section (paragraph 3.5.):

Notably, there was no overexpression of PD-L1 and TGFB1 in keratinocytes following addition of doxycycline (see Supplementary Materials Figure S5)

Round 2

Reviewer 2 Report

The authors improved manuscript well. It is worth to publish this manuscript in Cells.